# Contrastive Learning of Medical Visual Representations from Paired Images and Text

## Abstract

Learning visual representations of medical images is core to medical image understanding but its progress has been held back by the small size of hand-labeled datasets. Existing work commonly relies on transferring weights from ImageNet pretraining, which is suboptimal due to drastically different image characteristics, or rule-based label extraction from the textual report data paired with medical images, which is inaccurate and hard to generalize. We propose an alternative unsupervised strategy to learn medical visual representations directly from the naturally occurring pairing of images and textual data. Our method of pretraining medical image encoders with the paired text data via a bidirectional contrastive objective between the two modalities is domain-agnostic, and requires no additional expert input. We test our method by transferring our pretrained weights to 4 medical image classification tasks and 2 zero-shot retrieval tasks, and show that our method leads to image representations that considerably outperform strong baselines in most settings. Notably, in all 4 classification tasks, our method requires only 10% as much labeled training data as an ImageNet initialized counterpart to achieve better or comparable performance, demonstrating superior data efficiency.

## 1 Introduction

Medical image understanding has the potential to transform healthcare and has seen rapid progress with the use of deep neural architectures (Gulshan et al., 2016; Esteva et al., 2017; De Fauw et al., 2018; Rajpurkar et al., 2018b). Yet, with expert-level performance achieved only in some specialties and under some circumstances, medical image understanding remains a difficult task for the majority of specialties, mainly due to its challenging nature and the extreme scarcity of annotated data.

Existing work has followed two general approaches to obtain annotations for medical imaging tasks. The first approach has been using high-quality annotations created by medical experts (Abràmoff et al., 2016; Gulshan et al., 2016; Shih et al., 2019; Wang & Wong, 2020). However, the high cost of this approach has resulted in datasets that are mostly orders of magnitude smaller than natural image datasets such as ImageNet (Russakovsky et al., 2015). To remedy this, existing work has relied heavily on transferring model weights from ImageNet pretraining (Wang et al., 2017; Esteva et al., 2017; Irvin et al., 2019). This approach is suboptimal because, as shown in Figure 1, medical image understanding often requires representations of very fine-grained visual features that are drastically different from those required for identifying objects in natural images. As a result, Raghu et al. (2019) found that ImageNet pretraining often provides little to no benefit compared to simple random initialization.

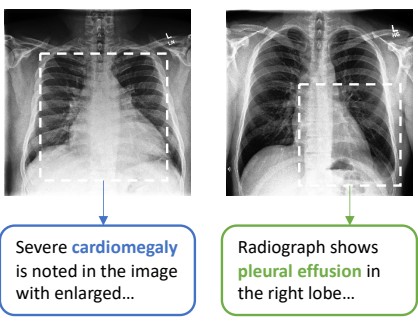

Figure 1: Two example chest radiograph images with different abnormality categories, along with sentences from their paired textual report and example views indicative of their characteristics.

A second popular approach is to use expert-crafted rules to extract labels from the textual reports accompanying the medical images. This approach has led to datasets of larger scale, since the text data paired with medical images are often produced naturally by medical experts in their routine work-

flow and abundant in a typical hospital's IT systems. Nevertheless, this rule-based label extraction approach has two limitations: 1) the rules are often inaccurate and limited to a few major categories (Wang et al., 2017), leading to very inefficient use of the textual report data; 2) these rules are often domain-specific and sensitive to the style of the text, making cross-domain and cross-institution generalization difficult (Irvin et al., 2019).

In efforts to make more efficient use of unlabeled image data, several recent studies have shown promising results from contrastive representation learning from natural images (Chen et al., 2020a; He et al., 2020; Grill et al., 2020). However, as we will show, applying these image view-based contrastive methods to medical images provides only marginal benefits compared to ImageNet pretraining, a result mostly due to the high inter-class similarity of the medical images as in Figure 1.

In this work, we aim to improve visual representations of medical images by combining the benefits of both learning from abundant textual data and unsupervised statistical approaches. We present *Contrastive VIsual Representation Learning from Text (ConVIRT)*, a framework for learning visual representations by exploiting the naturally occurring pairing of images and textual data. ConVIRT improves visual representations by maximizing the agreement between true image-text pairs versus random pairs via a bidirectional contrastive objective between the image and text modalities. We apply ConVIRT to the pretraining of medical image encoders, and show that it leads to higher-quality in-domain image representations that capture the subtlety of visual features required for medical image understanding tasks.

Compared to existing methods, ConVIRT has the advantages of utilizing the paired text data in a way agnostic to the medical specialty and requiring no additional expert input. This allows us to evaluate ConVIRT by transferring our pretrained weights to 4 different medical image classification tasks covering 2 different specialties. We find that the resulting models outperform all baseline initialization approaches, including the standard ImageNet pretraining and several strong baselines that also utilize the paired text data. Most notably, in all 4 tasks, ConVIRT requires only 10% as much labeled training data as an ImageNet initialized counterpart to achieve better or comparable performance. We further evaluate ConVIRT on two new zero-shot retrieval tasks, an image-image and a text-image retrieval task, and also find it superior to all baselines. To facilitate future research, we will make our code and the collected retrieval datasets available.

## 2 METHOD

### 2.1 TASK DEFINITION

We start by giving a formal description of our representation learning setting. We assume paired input $(\mathbf{x}_v, \mathbf{x}_u)$ where $\mathbf{x}_v$ represents one or a group of images, and $\mathbf{x}_u$ represents a text sequence which describes the imaging information in $\mathbf{x}_v$. Our goal is to learn a parameterized image encoder function $f_v$, which maps an image to a fixed-dimensional vector. We are then interested in transferring the learned image encoder function $f_v$ into downstream tasks, such as classification or image retrieval. In this work, we model the encoder function $f_v$ as a convolutional neural network (CNN).

We note that paired image-text data $(\mathbf{x}_v, \mathbf{x}_u)$ naturally exists for many medical domains. Medical experts such as radiologists produce textual descriptions of images as part of their routine workflow, some of which are also made publicly available (Demner-Fushman et al., 2016; Johnson et al., 2019).

### 2.2 CONTRASTIVE VISUAL REPRESENTATION LEARNING FROM TEXT

An overview of our method, ConVIRT, for learning $f_v$ is shown in Figure 2. At a high level, our method converts each input image $\mathbf{x}_v$ and text $\mathbf{x}_u$ into $d$-dimensional vector representations $\mathbf{v}$ and $\mathbf{u}$ respectively, following a similar processing pipeline. For each input image $\mathbf{x}_v$, our method starts by drawing a random view $\tilde{\mathbf{x}}_v$ from $\mathbf{x}_v$ with a sampled transformation function $t_v \sim \mathcal{T}$, where $\mathcal{T}$ represents a family of stochastic image transformation functions described later. Next, the encoder function $f_v$ transforms $\tilde{\mathbf{x}}_v$ into a fixed-dimensional vector $\mathbf{h}_v$, followed by a non-linear projection function $g_v$ which further transforms $\mathbf{h}_v$ into vector $\mathbf{v}$:

$$\mathbf{v} = g_v(f_v(\tilde{\mathbf{x}}_v)), \tag{1}$$

where $\mathbf{v} \in \mathbb{R}^d$. Similarly, for each text input $\mathbf{x}_u$, we obtain a span $\tilde{\mathbf{x}}_u$ from it following a sampling function $t_u$, and then a text representation $\mathbf{u}$ with: $\mathbf{u} = g_u(f_u(\tilde{\mathbf{x}}_u))$, where $f_u$ is a text encoder,

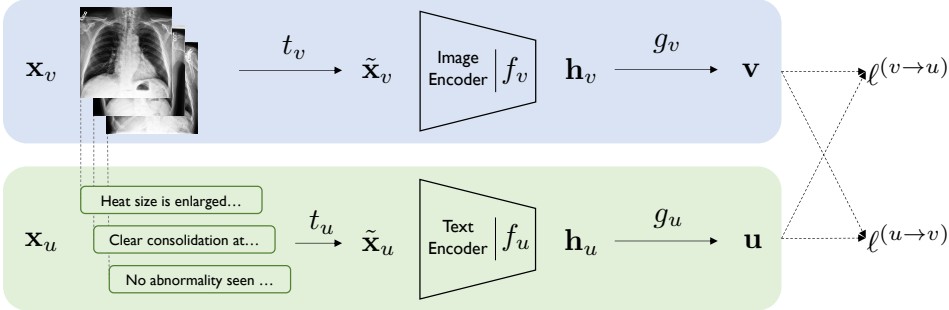

Figure 2: Overview of our ConVIRT framework. The blue and green shades represent the image and text encoding pipelines, respectively. Our method relies on maximizing the agreement between the true image-text representation pairs with bidirectional losses $\ell^{(v \to u)}$ and $\ell^{(u \to v)}$.

$g_u$ a projection, and $\mathbf{u} \in \mathbb{R}^d$. The projection functions $g_v$ and $g_u$ project representations for both modalities from their encoder space to the same $d$-dimensional space for contrastive learning.

At training time, we sample a minibatch of $N$ input pairs $(\mathbf{x}_v, \mathbf{x}_u)$ from training data, and calculate their representation pairs $(\mathbf{v}, \mathbf{u})$. We use $(\mathbf{v}_i, \mathbf{u}_i)$ to denote the $i$-th pair. The training objective of ConVIRT involves two loss functions. The first loss function is an image-to-text contrastive loss for the $i$-th pair:

$$\ell_i^{(v \to u)} = -\log \frac{\exp(\langle \mathbf{v}_i, \mathbf{u}_i \rangle / \tau)}{\sum_{k=1}^{N} \exp(\langle \mathbf{v}_i, \mathbf{u}_k \rangle / \tau)}, \tag{2}$$

where $\langle \mathbf{v}_i, \mathbf{u}_i \rangle$ represents the cosine similarity, i.e., $\langle \mathbf{v}, \mathbf{u} \rangle = \mathbf{v}^\top \mathbf{u} / \|\mathbf{v}\| \|\mathbf{u}\|$; and $\tau \in \mathbb{R}^+$ represents a temperature parameter. This loss takes the same form as the InfoNCE loss (Oord et al., 2018), and minimizing it leads to encoders that maximally preserve the mutual information between the true pairs under the representation functions. Intuitively, it is the log loss of an $N$-way classifier that tries to predict $(\mathbf{v}_i, \mathbf{u}_i)$ as the true pair. Note that unlike previous work which use a contrastive loss between inputs of the same modality (Chen et al., 2020a; He et al., 2020), our image-to-text contrastive loss is asymmetric for each input modality. We therefore define a similar text-to-image contrastive loss as:

$$\ell_i^{(u \to v)} = -\log \frac{\exp(\langle \mathbf{u}_i, \mathbf{v}_i \rangle / \tau)}{\sum_{k=1}^{N} \exp(\langle \mathbf{u}_i, \mathbf{v}_k \rangle / \tau)}. \tag{3}$$

Our final training loss is then computed as a weighted combination of the two losses averaged over all positive image-text pairs in each minibatch:

$$\mathcal{L} = \frac{1}{N} \sum_{i=1}^{N} \left( \lambda \ell_i^{(v \to u)} + (1 - \lambda) \ell_i^{(u \to v)} \right), \tag{4}$$

where $\lambda \in [0, 1]$ is a scalar weight.

## 2.3 REALIZATION

We note that our ConVIRT framework defined above is agnostic to the specific choice of image and text encoders, transformations and projection functions. In this work, following previous work (Chen et al., 2020a), we model $g_v$ and $g_u$ as separate learnable single-hidden-layer neural networks, i.e., $g_v(\cdot) = \mathbf{W}^{(2)} \sigma(\mathbf{W}^{(1)}(\cdot))$ where $\sigma$ is a ReLU non-linearity, and similarly for $g_u$.

For the image encoder $f_v$, we use the ResNet50 architecture (He et al., 2016) for all experiments, as it is the architecture of choice for much medical imaging work and is shown to achieve competitive performance. For the text encoder $f_u$, we use a BERT encoder (Devlin et al., 2019) followed by a max-pooling layer over all output vectors. We initialize our BERT encoder with the ClinicalBERT model (Alsentzer et al., 2019) pretrained on the MIMIC clinical notes, which achieved state-of-the-art performance on a suite of clinical NLP tasks. At training time we allow the encoder to adapt to our contrastive task by freezing the embeddings and the first 6 layers of this BERT encoder and fine-tuning the last 6 layers.

For the image transformation family $\mathcal{T}$ where $t_v$ is sampled from, we use sequential applications of five random transformations: *cropping*, *horizontal flipping*, *affine transformation*, *color jittering* and *Gaussian blur*. Different from recent work on contrastive visual representation learning (Chen et al., 2020a;b), we only apply brightness and contrast adjustments in *color jittering*, due to the monochrome nature of the medical images. For the text transformation function $t_u$, we apply a simple uniform sampling of a sentence from the input document $\mathbf{x}_u$ (i.e., $\tilde{\mathbf{x}}_u$ is a randomly sampled sentence from $\mathbf{x}_u$ for each minibatch). We did not use a more aggressive transformation mainly because sampling at the sentence level can preserve the semantic meaning of the sampled spans.

## 3    EXPERIMENTS

### 3.1    DATA FOR PRETRAINING

We test our ConVIRT framework by pretraining two separate image encoders covering different medical specialties using two separate paired image-text datasets:

- **Chest** image encoder: We use version 2 of the public **MIMIC-CXR** database (Johnson et al., 2019), which is a collection of chest radiograph images paired with their textual reports, and since its release has become a standard resource for studying multi-modal modeling of medical images. After preprocessing, this dataset contains a total of about 217k image-text pairs, with each pair containing an average of 1.7 images and 6.0 sentences.

- **Bony** image encoder: We obtain a collection of musculoskeletal image-text pairs from the Rhode Island Hospital system. Following chest images, musculoskeletal images constitute the second most common type of radiograph images in a typical hospital. This dataset contains a total of 48k image-text pairs, with each pair containing an average of 2.5 images and 8.0 sentences.

We include model implementation and pretraining details in Appendix A.

### 3.2    EVALUATION TASKS & DATA

We evaluate our pretrained image encoders on three downstream medical imaging tasks: image classification, zero-shot image-image retrieval and zero-shot text-image retrieval.

**Image Classification.**    We evaluate our pretrained image representations on four representative medical image classification tasks: 1) **RSNA Pneumonia Detection** (Wang et al., 2017; Shih et al., 2019), which involves binary classification of a chest radiograph image into either a *pneumonia* or a *normal* category; 2) **CheXpert** image classification (Irvin et al., 2019), which involves multi-label binary classification of a chest image for five individual labels, i.e., *atelectasis*, *cardiomegaly*, *consolidation*, *edema* and *pleural effusion*; 3) **COVIDx** image classification (Wang & Wong, 2020), which involves multi-class classification of a chest image into one of *COVID19*, *non-COVID pneumonia* or *normal* categories; and 4) **MURA** bony abnormality detection (Rajpurkar et al., 2018a), which involves binary classification of a musculoskeletal image into *abnormal* or *normal*. We report test accuracy for COVIDx given its balanced test set, and report the standard area under the receiver operating characteristic curve (AUC) metric for other tasks following previous work.

Following previous work (Hénaff et al., 2020; Chen et al., 2020a; He et al., 2020), for all tasks, we evaluate each pretrained image encoder under two individual settings: a **linear classification** setting, where the pretrained CNN weights are frozen and only a linear classification head is trained for the task; and a **fine-tuning** setting, where both the CNN weights and the linear head are fine-tuned. The two settings complement each other for evaluation purposes: while the linear setting directly evaluates the quality of the extracted image features with the pretrained CNN, the fine-tuning setting more closely resembles how the pretrained CNN weights are used in practical applications.

To further compare the **data efficiency** of different pretraining methods, for each setting we evaluate the image encoders with **1%**, **10%** and **all** training data, respectively (except for the COVIDx dataset where we omit the 1% setting due to the scarcity of data for some categories). To control the variance in results, for all settings and models, we report average results aggregated over 5 independent training runs. We include further dataset processing and model training details in Appendix B.

**Zero-shot Image-image Retrieval.** This evaluation is similar to the conventional content-based image retrieval setting in which we search for images of a particular category using a representative *query* image. For evaluation, a group of query images and a larger collection of *candidate* images, each with a categorical label, are given to a pretrained CNN encoder. We encode each query and candidate image with this encoder, and then for each query, rank all candidates by their cosine similarities to the query in descending order. Since a widely-used annotated benchmark for this setting is not available, we create our own dataset by re-using existing annotations in the CheXpert dataset (Irvin et al., 2019) and additional expert annotations from a board-certified radiologist. The resulting dataset covers 8 different chest abnormality categories, each with 10 expert-annotated query and 200 candidate images. We include the detailed collection and annotation procedure in Appendix C, and refer to this dataset as **CheXpert 8×200 Retrieval Dataset**. We focus our evaluation on retrieval precision, and evaluate our models with Precision@$k$ metrics where $k = 5, 10, 100$.

**Zero-shot Text-image Retrieval.** This setting is similar to the image-image retrieval setting, but instead of using query images, we retrieve images of a particular category with textual queries. For this purpose, we ask a radiologist to write 5 diverse and representative textual descriptions for each of the 8 abnormality categories for the same CheXpert 8x200 candidate images (see Appendix D for details). At test time, for each query we encode its text with the learned text encoder $f_u$ and then retrieve from candidate images in a similar way. This evaluation not only evaluates the quality of the learned image representations, but also the alignment between the text representations and the image representations. We again use Precision@$k$ metrics where $k = 5, 10, 100$.

### 3.3 BASELINE METHODS

We compare ConVIRT against the following standard or competitive initialization methods:

- **Random Init.**: For all tasks we initialize the ResNet50 with its default random initialization.
- **ImageNet Init.**: We use CNN weights pretrained on ImageNet (Russakovsky et al., 2015), which remains a dominant initialization approach for medical imaging work (Raghu et al., 2019).
- **Caption-LSTM**: We initialize the CNN weights with ImageNet, and then pretrain it with an image captioning task using the standard CNN-LSTM with attention architecture (Xu et al., 2015). For the captioning task, we train the model to decode the paired textual report from the encoded image representations. Compared to the random or ImageNet initializations, this is an "in-domain" initialization baseline which uses the paired text data for representation learning.
- **Caption-Transformer**: In this initialization we replace the CNN-LSTM model in Caption-LSTM with a CNN-Transformer-based captioning model in Cornia et al. (2020), which recently achieves state-of-the-art results on the COCO image captioning benchmark (Lin et al., 2014).
- **Contrastive-Binary**: This baseline differs from our method by contrasting the paired image and text representations with a binary classification head, as is widely done in visual-linguistic pre-training work (Tan & Bansal, 2019; Su et al., 2020). For each input pair, we first project encoder outputs $\mathbf{h}_v$ and $\mathbf{h}_u$ into the same dimension with linear layers, concatenate them, and use a MLP network to predict a binary probability of whether the input is a real or a "fake" pair, which we train with a binary cross-entropy loss. During training, for each $(\mathbf{x}_v, \mathbf{x}_u)$ pair in the training set, we construct a "fake" pair by replacing $\mathbf{x}_u$ with a randomly sampled one from the dataset. We expect that the binary classification task requires the encoder to learn reasonable representations of the input images, and therefore is a stronger in-domain initialization baseline.

For fair comparison, for all baselines that require paired image-text data, we use the same datasets as in our contrastive pretraining. For the captioning-based methods, we always use the model checkpoints that achieve the best CIDEr score (Vedantam et al., 2015) on a held-out validation set.

## 4 RESULTS

### 4.1 CLASSIFICATION TASKS

**Linear Classification.** We present all linear classification results for the medical imaging tasks in Table 1a. We find that compared to random initialization, ImageNet initialization provides markedly

Table 1: Results for the medical image classification tasks: (a) linear classification; (b) fine-tuning setting. All results are averaged over 5 independent models. Best results for each setting are in boldface. COVIDx 1% setting is omitted due to the scarcity of labels in COVIDx.

(a)

| Method | RSNA (AUC) | | | CheXpert (AUC) | | | COVIDx (Accu.) | | MURA (AUC) | | |
|---|---|---|---|---|---|---|---|---|---|---|---|
| | 1% | 10% | all | 1% | 10% | all | 10% | all | 1% | 10% | all |
| *General initialization methods* | | | | | | | | | | | |
| Random Init. | 55.0 | 67.3 | 72.3 | 58.2 | 63.7 | 66.2 | 69.2 | 73.5 | 50.9 | 56.8 | 62.0 |
| ImageNet Init. | 82.8 | 85.4 | 86.9 | 75.7 | 79.7 | 81.0 | 83.7 | 88.6 | 63.8 | 74.1 | 79.0 |
| *In-domain initialization methods* | | | | | | | | | | | |
| Caption-Transformer | 84.8 | 87.5 | 89.5 | 77.2 | 82.6 | 83.9 | 80.0 | 89.0 | 66.5 | 76.3 | 81.8 |
| Caption-LSTM | 89.8 | 90.8 | 91.3 | 85.2 | 85.3 | 86.2 | 84.5 | **91.7** | 75.2 | 81.5 | 84.1 |
| Contrastive-Binary | 88.9 | 90.5 | 90.8 | 84.5 | 85.6 | 85.8 | 80.5 | 90.8 | 76.8 | 81.7 | 85.3 |
| ConVIRT (Ours) | **90.7** | **91.7** | **92.1** | **85.9** | **86.8** | **87.3** | **85.9** | **91.7** | **81.2** | **85.1** | **87.6** |

(b)

| Method | RSNA (AUC) | | | CheXpert (AUC) | | | COVIDx (Accu.) | | MURA (AUC) | | |
|---|---|---|---|---|---|---|---|---|---|---|---|
| | 1% | 10% | all | 1% | 10% | all | 10% | all | 1% | 10% | all |
| *General initialization methods* | | | | | | | | | | | |
| Random Init. | 71.9 | 82.2 | 88.5 | 70.4 | 81.1 | 85.8 | 75.4 | 87.7 | 56.8 | 61.6 | 79.1 |
| ImageNet Init. | 83.1 | 87.3 | 90.8 | 80.1 | 84.8 | 87.6 | 84.4 | 90.3 | 72.1 | 81.8 | 87.0 |
| *In-domain initialization methods* | | | | | | | | | | | |
| Caption-Transformer | 86.3 | 89.2 | 92.1 | 81.5 | 86.4 | **88.2** | 88.3 | 92.3 | 75.2 | 83.2 | 87.6 |
| Caption-LSTM | 87.2 | 88.0 | 91.0 | 83.5 | 85.8 | 87.8 | 83.8 | 90.8 | 78.7 | 83.3 | 87.8 |
| Contrastive-Binary | 87.7 | 89.9 | 91.2 | 86.2 | 86.1 | 87.7 | 89.5 | 90.5 | 80.6 | 84.0 | 88.4 |
| ConVIRT (Ours) | **88.8** | **91.5** | **92.7** | **87.0** | **88.1** | 88.1 | **90.3** | **92.4** | **81.3** | **86.5** | **89.0** |

better representations, despite pretrained on a very different domain of images; in-domain image initialization methods that use paired image-text data further improve over ImageNet initialization in almost all settings. Among the in-domain initialization methods, our proposed ConVIRT pretraining achieves the best overall results in all settings. Notably, we find on three out of the four tasks, with only 1% training data ConVIRT is able to achieve classification results better than the default ImageNet initialization with 100% training data, highlighting the high quality of the learned representations from ConVIRT.

**Fine-tuning.** We show the fine-tuning evaluation results in Table 1b. Similar to the linear setting, we find that: 1) ImageNet initialization is again better than random initialization with smaller margins; 2) all in-domain initialization methods are better than the popular ImageNet initialization in most settings; and 3) our proposed ConVIRT pretraining again achieves the best overall results in 10 out of the 11 settings, with the exception of the CheXpert dataset with all training data used, where the result of ConVIRT is similar to that of the Caption-Transformer result. Most notably, on all datasets, with only 10% labeled training data ConVIRT achieves classification results that are better or close to the ImageNet initialization with 100% training data results.

We also notice that our conclusion of using ImageNet versus random initialization is different from Raghu et al. (2019): while they showed comparable results from the two strategies, we find that using ImageNet initialization is still superior than random initialization in most results, justifying its popularity. Upon closer examination, we conjecture that this is likely due to under-optimization of their models: while our ResNet50 with random initialization achieves an average AUC of 85.8 on the CheXpert dataset, their ResNet50 model only achieved 83.5 AUC on the same evaluation set.

## 4.2 RETRIEVAL TASKS

We present the zero-shot image-image and text-image retrieval results in Table 2. For the image-image retrieval setting, we present additional results from fine-tuning our pretrained model on all CheXpert training data, and use them as "upper bounds" of the results obtained from the use of supervised labels. We find that: 1) using ImageNet pretrained CNN weights in a zero-shot image retrieval setting is only better than random guess by small margins; 2) all in-domain pretrained CNN

Table 2: Zero-shot image-image and text-image retrieval results on the CheXpert $8\times200$ datasets. *Random* shows results from a random guess; *ConVIRT + CheXpert Supervised* shows results from further fine-tuning the pretrained weights with supervised training data. Text-image retrieval results are not obtained for some methods due to the lack of text encoders.

| | Image-Image Retrieval | | | Text-Image Retrieval | | |
| Method | Prec@5 | Prec@10 | Prec@50 | Prec@5 | Prec@10 | Prec@50 |
|---|---|---|---|---|---|---|
| Random | 12.5 | 12.5 | 12.5 | 12.5 | 12.5 | 12.5 |
| ImageNet | 14.8 | 14.4 | 15.0 | – | – | – |
| *In-domain initialization methods* | | | | | | |
| Caption-Transformer | 29.8 | 28.0 | 23.0 | – | – | – |
| Caption-LSTM | 34.8 | 32.9 | 28.1 | – | – | – |
| Contrastive-Binary | 38.8 | 36.6 | 29.7 | 15.5 | 14.5 | 13.7 |
| ConVIRT (Ours) | **45.0** | **42.9** | **35.7** | **60.0** | **57.5** | **48.8** |
| *Fine-tuned* | | | | | | |
| ConVIRT + CheXpert Supervised | 56.8 | 56.3 | 48.9 | – | – | – |

weights achieve much better retrieval performance than ImageNet weights; and 3) our proposed ConVIRT pretraining achieves the best overall retrieval results on all metrics. We find that while Contrastive-Binary performs notably better than other baselines in the image-image retrieval setting, its text-image retrieval results are far from ConVIRT pretraining. We conjecture that the lack of an explicit similarity-based loss function in the Contrastive-Binary baseline results in misaligned representations in the image and text space, leading to poor results in text-image retrieval.

To understand how well ConVIRT pretraining helps separate images from different abnormality categories in its encoding space, in Figure 3 we present t-SNE plots (Maaten & Hinton, 2008) of candidate images in the CheXpert 8x200 dataset for five selected categories, from the ImageNet pretrained CNN encoder and the ConVIRT pretrained encoder. It is worth noting that clustering images in our setting is much more challenging than that in the general object classification setting due to the high inter-class similarity of the medical images. Nevertheless we find that ConVIRT pretraining achieves a better clus-

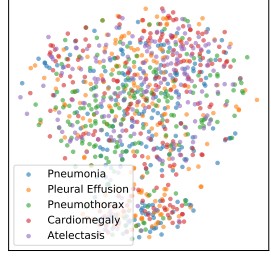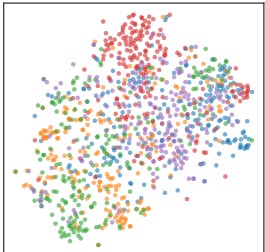

(a) ImageNet Pretraining  (b) ConVIRT Pretraining

Figure 3: t-SNE visualizations of encoded image representations from different pretraining methods.

tering of the images in the t-SNE plots. On the other hand, the lack of clear separations between groups suggests room for further improvement.

## 5 ANALYSIS AND DISCUSSION

**Comparisons to Image-only Contrastive Learning.** ConVIRT shows superior results against baselines in evaluation, but an important question remains as to how it compares against existing image-only contrastive visual representation learning methods. We study this by running two popular such methods, SimCLR (Chen et al., 2020a) and MoCo v2 (Chen et al., 2020b), on the same collection of images that we used in our pretraining. We present the results in Table 3 and include model training details in Appendix E. We find that compared to ImageNet initialization, both contrastive methods lead to marginal to moderate improvements on the classification

Table 3: Comparisons of ConVIRT to image-only unsupervised image representation learning approaches.

| Method | RSNA Linear (1%, AUC) | CheXpert Linear (1%, AUC) | Image-Image (Prec@10) |
|---|---|---|---|
| ImageNet | 82.8 | 75.7 | 14.4 |
| SimCLR | 86.3 | 77.4 | 17.6 |
| MoCo v2 | 86.6 | 81.3 | 20.6 |
| ConVIRT | 90.7 | 85.9 | 42.9 |

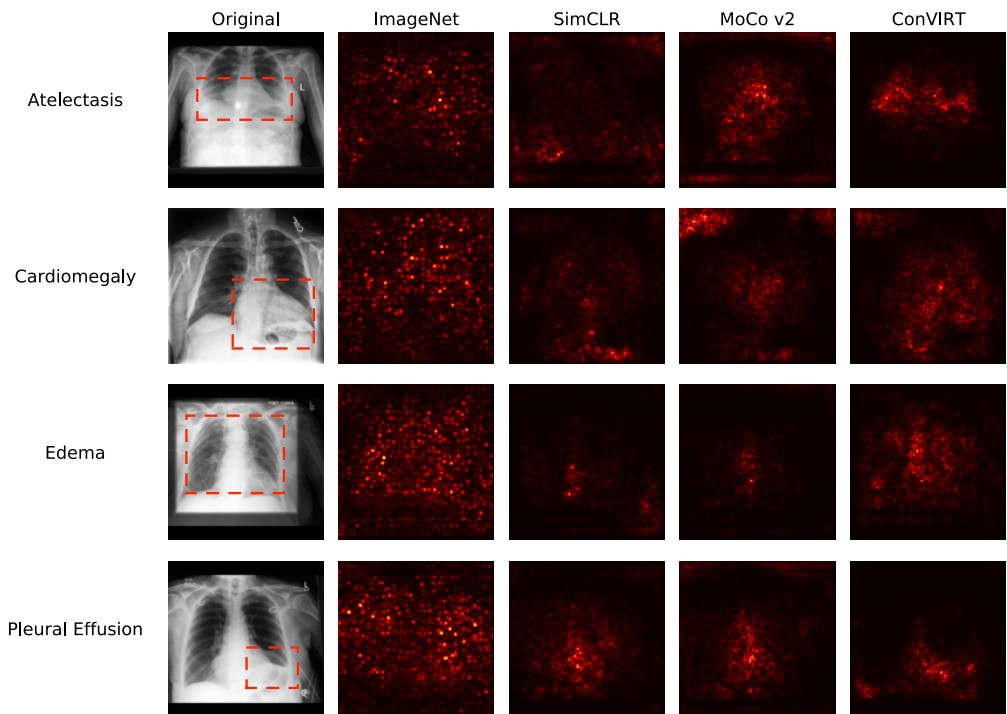

Figure 4: Saliency maps on sampled images for 4 abnormality categories in the CheXpert dataset. For each image we present maps for ImageNet, SimCLR, MoCo v2 and our ConVIRT initializations. Ground truth regions that are indicative of the abnormalities are shown as red boxes in the images.

and retrieval tasks. However, our training strategy substantially outperforms both methods on all tasks, demonstrating its effective use of information from the paired text data.

To understand the representational difference that has led to this difference in performance, for all four initialization methods, we visualize in Figure 4 the saliency maps (Simonyan et al., 2014) corresponding to the correct class on sampled images from the CheXpert dataset. Models for all initialization methods are trained with 1% CheXpert training data under the linear classification setting (with pretrained CNN weights frozen). We find that ImageNet pretraining has led to models that focus on trivial visual features that are mostly irrelevant to the task, and that the model with ConVIRT pretrained weights has focused on much more relevant areas than those with SimCLR and MoCo v2 pretraining, suggesting more effective representation learning. For example, for *atelectasis*, while the ConVIRT model has correctly focused on the bottom of the lung regions, the SimCLR model has much more scattered focus and the MoCo model has incorrectly focused on the heart region.

**Correlation Between Contrastive Loss and End Task Performance.** To understand the relation between a model's performance on the ConVIRT pretraining task and its performance on the downstream tasks, we ran an analysis where for every 5 epochs during the pretraining, we transferred the pretrained checkpoint to the downstream tasks and evaluate its performance. The pretraining was run for a total of 200 epochs, and 40 points were obtained with varying validation loss and end task results. Figure 5 presents the results of the models' validation loss on the pretraining task, and its achieved performance on the RSNA 1% data linear evaluation and the two retrieval tasks. For all three tasks, we find a clear positive correlation between the pretraining performance and the end task performance. This corroborates that by learning with the ConVIRT objective, the image encoder learns gradually improved representations for the end tasks, and suggests that further improvement on the pretraining task may have positive impact on the end task performance.

**Hyperparameter Analysis.** We run experiments to study the impact of hyperparameters, and find that: 1) similar to previous work on image-only contrastive learning (Chen et al., 2020a), the pretraining results are most sensitive to the choice of the temperature value $\tau$; 2) unlike previous work,

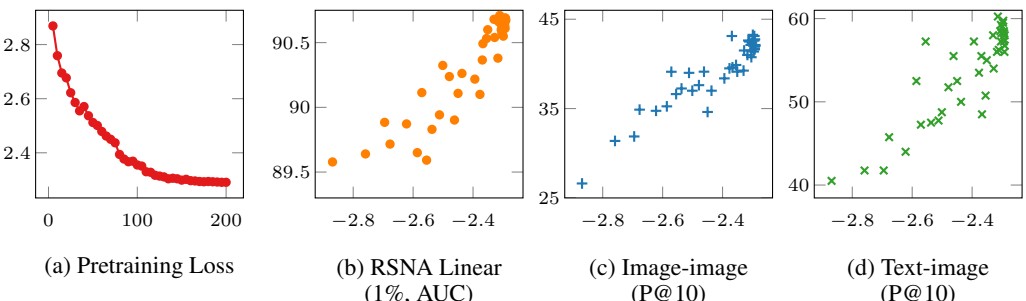

Figure 5: (a) shows pretraining validation loss at different epochs; (b)-(d) shows correlation between the pretraining loss and the performance of three end tasks. For (a) the x-axis shows the training epoch number, and for (b)-(d) the x-axis shows the negative value of the pretraining loss (i.e., $-\mathcal{L}$) on a held-out validation set.

changing batch size does not lead to substantial change in the classification results; and 3) using linear projection heads instead of non-linear ones notably hurts the retrieval results. We include our detailed comparisons in Appendix F.

## 6 RELATED WORK

Our work is most relevant to existing work on medical image classification, which we have covered in Section 1, and textual report generation from medical images (Wang et al., 2018; Jing et al., 2018; Liu et al., 2019). A dominant approach for initializing medical image encoders in this work has been using encoder weights pretrained on ImageNet, despite the drastic difference in image characteristics (Raghu et al., 2019). Instead, our work proposes an alternative in-domain pretraining strategy, and compares ImageNet pretraining with different pretraining approaches that also use the paired text data. To our knowledge our work represents the first systematic attempt in this direction.

Our work is inspired by the recent line of work on image view-based contrastive visual representation learning (Hénaff et al., 2020; Chen et al., 2020a; He et al., 2020; Grill et al., 2020), but differs from existing studies by the contrastive learning with text modality, which as we show in Section 5, is more effective in learning high-quality representations of medical images.

Another line of work related to ours is visual-linguistic representation learning (Lu et al., 2019; Tan & Bansal, 2019; Su et al., 2020). Among existing studies, Ilharco et al. (2020) and Gupta et al. (2020) used a cross-modality contrastive objective related to ours, but for the purpose of probing visual-linguistic models and learning phrase grounding, respectively. Our work differs from this line of work in several crucial ways: 1) existing work in visual-linguistic learning focused on learning visual representations from paired text via a binary contrastive prediction task, whereas we showed the superior performance of the new cross-modality NCE objectives in our setting; 2) existing work has primarily used object representations extracted from image segmentation models in their preprocessing steps, making them less applicable to medical image understanding tasks where anatomical segmentations are extremely hard to obtain; 3) while existing work has run evaluation primarily on visual-linguistic tasks such as visual question answering, we instead focus on evaluation with classification and retrieval tasks which are at the center of medical image understanding research.

## 7 CONCLUSION

We presented ConVIRT, an unsupervised method for learning medical visual representations from naturally occurring pairing of images and text. Our method relies on contrasting the image representations with the paired text data via a bidirectional objective between the two modalities. On 4 medical image classification tasks and 2 image retrieval tasks, ConVIRT outperformed other strong in-domain initialization methods that also use the text data, and led to representations with notably higher quality. Compared to ImageNet pretraining, ConVIRT is able to achieve the same level of classification accuracy with an order of magnitude less labeled data. We hope that our work can inspire future work that makes more efficient use of textual data for medical image understanding.

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

## A  MODEL IMPLEMENTATION AND PRETRAINING DETAILS

**Dataset Preprocessing.**   For the MIMIC-CXR chest radiograph dataset, we use the publicly available JPG version of it.[1] For both the MIMIC-CXR chest dataset and the Rhode Island Hospital bone image datasets, we resize the image files to have a size of 256 on the larger side. For the textual radiology report data, we first tokenize all reports with the default English tokenizer in version 4.0.0 of the CoreNLP library (Manning et al., 2014). Next, we keep only the *Findings* and *Impression* sections and remove all other sections. We remove all image-text pairings from the dataset where the text section is empty or has less than 3 tokens. This preprocessing procedure gives us about 217k total image-text pairs for pretraining our chest image encoder and 48k total pairs for pretraining our bone image encoder.

**Image and Text Encoders.**   For the image encoder, we use the standard ResNet50 implementation provided by the torchvision library. For the text encoder, we use the BERT base encoder offered by the Transformers library (Wolf et al., 2019) and initialize it with the ClinicalBERT model (Alsentzer et al., 2019) pretrained on the MIMIC clinical notes. We also experimented with training a specialized BERT encoder on a large collection of radiology notes but found that it made no substantial difference in the pretraining results. At pretraining time we freeze the embeddings and the first 6 layers of this BERT encoder, and only fine-tune the last 6 layers for our contrastive task.

**Other Hyperparameters.**   For contrastive learning, we use projection layers with an output dimension $d = 512$, a temperature value $\tau = 0.1$, a loss weight $\lambda = 0.75$. These hyperparameter settings are obtained by comparing the linear evaluation validation scores on the RSNA image classification task with the pretrained ResNet50 weights. For the image transformation family $\mathcal{T}$, we adopt the implementations offered by the torchvision library.[2] We apply *random cropping* with a ratio sampled from $[0.6, 1.0]$; *horizontal flipping* with $p = 0.5$; *affine transformation* with a degree sampled from $[-20, 20]$, max horizontal and vertical translation fractions of 0.1, and a scaling factor sampled from $[0.95, 1.05]$; *color jittering* with brightness and contrast adjustment ratios sampled from $[0.6, 1.4]$; and *Gaussian blur* with $\sigma \in [0.1, 3.0]$. All images are resized to $224 \times 224$ after the transformation $t_v$ is applied. Limited by computational resources, we arrive at these image transformation parameters via preliminary experiments rather than a systematic search.

**Pretraining Details.**   At pretraining time, for each dataset, we randomly sample 5k image-text pairs to form a held-out validation set. We we use the Adam optimizer (Kingma & Ba, 2015) with an initial learning rate of 1e-4 and weight decay of 1e-6. We initialize the image encoder with ImageNet pretrained weights at the beginning of pretraining, and use a fixed batch size of 32. We calculate the validation loss every 5000 steps, and if the validation loss does not decrease after 5 straight evaluation runs, we anneal the learning rate by a factor of 0.5. We stop pretraining after 200 evaluation runs, and save the model checkpoint that achieves the lowest validation loss. For efficiency, we employ mixed-precision training, and for reference, the whole pretraining run on the MIMIC-CXR dataset took about 3 days on a single Titan RTX GPU card.

## B  IMAGE CLASSIFICATION EXPERIMENTS

We prepared and used the 4 image classification datasets following the procedures below:

1. **RSNA Pneumonia Detection** (Wang et al., 2017; Shih et al., 2019): we used the original version of this dataset available at its Kaggle page,[3] which contains 25184/1500/3000 annotated images in its training/validation/test sets, respectively.

2. **CheXpert** image classification (Irvin et al., 2019): we downloaded the original version of this dataset from its official website.[4] Since the original expert-labeled test set of this dataset is hidden and not included as part of the release, we instead followed Raghu et al. (2019) and used the original expert-labeled validation set as our test set, and randomly sampled 5000 images from

---

[1]https://physionet.org/content/mimic-cxr-jpg/2.0.0/

[2]https://github.com/pytorch/vision

[3]https://www.kaggle.com/c/rsna-pneumonia-detection-challenge

[4]https://stanfordmlgroup.github.io/competitions/chexpert/

the original training set for validation purpose. The resulting dataset contains 218414/5000/234 images in each split.

3. **COVIDx** image classification (Wang & Wong, 2020): we prepared this dataset following the scripts provided by its authors.[5] We used the version 4 of this dataset, the latest version at the time of this work. We additionally randomly sampled 300 images from the training set for validation, resulting in a dataset with 13598/300/300 images in each split.

4. **MURA** bony abnormality detection (Rajpurkar et al., 2018a): we downloaded the original version of this dataset from its website.[6] Similar to the CheXpert dataset, we again used the original validation set as our test set, and randomly sampled 10% images from the training set for validation, resulting in a dataset with 33078/3730/3197 images in each split. Different from the other 3 datasets, the MURA dataset uses patient-level evaluation, meaning that the prediction results from different images of the same patient needs to be aggregated to produce a final prediction for the patient, which is then scored against the gold patient label. We therefore followed Rajpurkar et al. (2018a) and at test time aggregated result for a patient by averaging the predicted probabilities from multiple images.

**Classification Model Training Details.** For all models that require ImageNet pretrained initialization, we use the pretrained weights from torchvision, which achieves an ImageNet top-5 error rate of 7.13%. For all datasets, we first zero-pad the input image to be square, and then resize it to be $224 \times 224$. For training, we use the Adam optimizer with an initial learning rate of 1e-3 for the COVIDx task and 1e-4 for the other three tasks. We additionally apply a weight decay of 1e-6 and a dropout before the last classification layer with $p = 0.2$ in all tasks. All classification models are trained with a batch size of 64. In the fine-tuning evaluation setting, we first "warmup" the classification head by freezing the CNN weights and only training the classification head with a learning rate of 1e-3 for 200 steps, after which we unfreeze the CNN weights and fine-tune the entire network together. Validation score is obtained after each epoch of training and we anneal the learning rate by a factor of 0.5 if the validation score is not improved after 3 epochs. The training is stopped after no validation improvement is observed for 10 straight epochs, at which point the model checkpoint with the highest validation score is evaluated on the test set.

## C    IMAGE-IMAGE RETRIEVAL DATASET COLLECTION

We create the CheXpert $8 \times 200$ Retrieval Dataset with 8 different abnormality categories commonly found in Chest radiograph images, including *atelectasis*, *cardiomegaly*, *edema*, *fracture*, *pleural effusion*, *pneumonia*, *pneumothorax* and a special *no finding* category indicating that no obvious abnormality is found in the image. We create the dataset by reusing existing rule-labeled annotations in the CheXpert dataset (Irvin et al., 2019) and additional expert annotations. To create the candidate images for a category label $\ell$, we go through all images in the CheXpert training set, and keep an image as a candidate image if only its label for $\ell$ is positive and all other categories negative. We only include images with this "exclusive positivity" as candidate images, mainly to avoid confounding results between categories in retrieval evaluation.

To create the query images for a category $\ell$, we again first pre-select 50 exclusively positive images for this category in the CheXpert training set (with all candidate images excluded). Next, we ask a board-certified radiologist to examine each of the 50 images, and exclude images that: 1) might indicate additional abnormalities other than $\ell$, 2) have uncommon color or contrast distortions in the image, or 3) are not well posed during the capture of the image. This procedure is mainly to avoid including query images that have uncommon features and may therefore bias the retrieval evaluation results. At the end, we aggregate the annotation results from the radiologist and keep 10 query images for each abnormality category.

## D    TEXT-IMAGE RETRIEVAL DATASET COLLECTION

For the text-image retrieval dataset, we first reuse all candidate images from the CheXpert $8 \times 200$ image-image retrieval dataset described above, with 200 images for each of 8 categories. To create

---

[5] https://github.com/lindawangg/COVID-Net
[6] https://stanfordmlgroup.github.io/competitions/mura/

Table 4: Example textual queries for each of the 8 categories in the text-image retrieval task.

| Image Category | Example Textual Query |
|---|---|
| Atelectasis | Platelike opacity likely represents atelectasis. |
| Cardiomegaly | The cardiac silhouette is enlarged. |
| Edema | The presence of hazy opacity suggests interstitial pulmonary edema. |
| Fracture | A cortical step off indicates the presence of a fracture. |
| Pleural Effusion | The pleural space is partially filled with fluid. |
| Pneumonia | A pulmonary opacity with ill defined borders likely represents pneumonia. |
| Pneumothorax | A medial pneumothorax is present adjacent to the heart. |
| No Finding | No clinically significant radiographic abnormalities. |

the textual queries for each abnormality category, we ask a board-certified radiologist to write at least 5 different sentences that he will use to describe this abnormality in radiology reporting. We additionally set the following requirements: 1) the sentences must describe the category with no ambiguity and must not include other categories; 2) the sentences must be diverse from each other; and 3) the sentences should not include very specific anatomic locations or rare clinical observations. At the end, we aggregate the results and keep 5 textual queries for each abnormality category. For reference, we present example textual queries in Table 4.

## E    EXPERIMENTS ON IMAGE-ONLY CONTRASTIVE LEARNING METHODS

We run experiments with two popular image-only contrastive visual representation learning methods: SimCLR (Chen et al., 2020a) and MoCo v2 (Chen et al., 2020b). For a fair comparison, in both experiments we use the exact same set of images from the MIMIC-CXR dataset that we use in the pretraining of our method and the baselines. Our settings for each method are:

- **SimCLR**: We use the open PyTorch implementation available at `https://github.com/sthalles/SimCLR`. For image encoder we use ResNet50. We use cosine similarity in the loss function, set the temperature value to 0.1 and set the output dimension to 128. We use the default image augmentation functions in the paper except for the *color jittering* transformation where we set the saturation and hue adjustment to 0 due to the monochrome nature of our medical images. For training, we use the Adam optimizer with an initial learning rate of 3e-4 and weight decay of 1e-4. We set batch size to 128 and run training on a single GPU card for 100 epochs, as we find that increasing the batch size or number of epochs does not lead to improved results. We use the default settings for all other parameters.

- **MoCo v2**: We use the authors' PyTorch implementation available at `https://github.com/facebookresearch/moco`. For image encoder we use ResNet50. We follow the default MoCo v2 setting and use a temperature value of 0.07 and an output dimension of 128. Similarly, we adopt the default image augmentation functions except for the *color jittering* transformation where we set the saturation and hue adjustment to 0. For training, we use the SGD optimizer with a learning rate of 0.0075 and weight decay of 1e-4. We use a batch size of 64 and a queue size of 4096, and run parallel training on two GPU cards for 100 epochs, as we find that further increasing the batch size or number of epochs does not lead to improved results. During training, we anneal the learning rate by a factor of 0.1 at the 60th and 80th epochs.

## F    HYPERPARAMETER ANALYSIS

Similar to previous work on unsupervised image representation learning (Chen et al., 2020a; He et al., 2020), we first find that the effectiveness of ConVIRT pretraining is most sensitive to the temperature value $\tau$. As shown in Table 5, using a temperature much lower than the ideal value ($\tau = 0.01$) hurts the retrieval results, and a temperature much larger ($\tau = 1$) notably hurts the performance on all tasks. Unlike previous work, we find that using a smaller or larger batch size hurts the retrieval performance, but neither setup brings substantial impact to the classification results. Lastly, we find that replacing the non-linear projection heads in $g_v$ and $g_u$ with linear layers hurts the

Table 5: Evaluation results with different hyperparameters, for the RSNA 1% data linear evaluation, image-image retrieval and text-image retrieval tasks. *bs* represents batch size and *linear proj.* represents using linear projection layers for $g_v$ and $g_u$. Our default model uses $\tau = 0.1$, bs $= 32$ and non-linear projections.

| Settings | RSNA Linear (1%, AUC) | Image-Image (Prec@10) | Text-Image (Prec@10) |
|---|---|---|---|
| ConVIRT (default) | 90.7 | 42.9 | 57.5 |
| $\tau = 0.01$ | 90.7 | 40.5 | 21.0 |
| $\tau = 1$ | 89.6 | 25.0 | 31.0 |
| bs = 16 | 90.3 | 40.0 | 55.8 |
| bs =128 | 90.3 | 39.3 | 50.3 |
| linear proj. | 90.6 | 40.8 | 55.8 |

retrieval results moderately, suggesting worse representations. However, this is again not reflected notably in the RSNA classification results.

