# OpenReview forum: "Contrastive Learning of Medical Visual Representations from Paired Images and Text"
_ICLR.cc/2021/Conference — Reject_

### Official Review · AnonReviewer4 · 2020-10-28
**Good performance but limited novelty.**

**Rating:** 4
**Confidence:** 4

**Review:**

Summary:

1. This paper tackles the medical image understanding problem. The aim of this paper is to learn a generic feature representation for medical image that could benefits downstream tasks like medical image classification, zero-shot classification. The main contribution of this paper is proposing a contrastive loss that the matched pair of image and text should have a higher corresponding score than the mis-matched pairs.

Pros:
1. The problem of learning a generic representation from limited training data is important.
2. Comparing with the baseline, the performance of the proposed approach looks good.
3. The paper is clearly written.

Weakness:
1. My first concern is the novelty. If I understand correctly, the main novelty of this paper is proposing the losses (Eq. 2 and Eq. 3) in the section of contrastive visual representation learning from text. However those losses are well-known in the domain of image-text retrieval. Current vision-language BERT (VLBERT) approaches use similar loss to optimize their model [Lu et. al., 2019]. The difference is that, in VLBERT papers, the model is trying to contrast the positive pair of image and text against all negative pair of image and texts that sampled from a batch: $$\ell = - \log \frac{\exp(<v_i, u_i> / \gamma)}{\sum_{k=1, k\neq i}^N\sum_{j=1, j\neq i}^N \exp(<v_j, u_k>/\gamma)}.$$ While the proposed approach uses Eq. 2 and Eq. 3 to optimize text retrieval and image retrieval, respectively. However, the motivation of separating the loss into Eq. 2 and Eq. 3 in unclear.
2. In the related work, recent advance in vision-language BERT has been mentioned (Last paragraph of Sect. 6 Related work). The paper stated that (in point 2) 'existing work has focused on visual-linguistic tasks such as visual question answering.', which might not be accurate. As current vision-language BERT models are trying to learn a generic representation for both image and text, where downstream tasks are ways to evaluate the representation. The aim of vision-language BERT is aligned with the aim of this paper.
3. The paper mentioned multiple times that the proposed approach is specific for medical images (Last paragraph of Related work). However this is unclear to me why the proposed approach is specific for medical images?

I would be happy to increase my rating, if the author could differentiate the proposed approach (losses) with the one used in current visual-language BERT. It would be the best if the author could state clearly why the proposed approach would be beneficial for medical images (Otherwise, 'medical' in the title of this paper might need to be removed). The author might also need to re-state the novelty contribution of this paper.

---

> ### Author Response · Authors · 2020-11-19
> **Authors' Response to Reviewer 3**
>
> Dear Reviewer 3,
>
> Thank you for your detailed comments, and for recognizing the importance of the addressed problem, the strong performance of our method, and the clear presentation of our paper! You are mainly concerned with the difference between our work and recent work on vision-language pretraining, and whether our method is specific to medical imaging. We clarify the differences between our work and recent work on vision-language pretraining in our **general response** to all reviewers, and address your detailed comments below:
>
> **Weakness 1** (difference with vision-language pretraining): Please see our **general response** to all reviewers in which we clarify the crucial differences between our submission and existing work.
>
> **Weakness 2** (inaccurate claim about visual-linguistic tasks): With hindsight, we agree that our claim in the original submission may be misleading. Learning good image representations is indeed a primary aim of vision-language pretraining work, and we will make it more accurate in an updated version of our paper. We however did hope to emphasize that, existing work such as ViLBERT [1] or VL-BERT [2] mainly focused their experiments on visual-linguistic tasks such as visual question answering or visual commonsense reasoning, while our experiments were mainly focused on image-only classification tasks, which are at the core of medical image understanding research.
>
> **Weakness 3** (specificity for medical images): We believe that this was also due to a misunderstanding. We intended to emphasize in our paper that our method was mainly motivated by the problem of learning fine-grained representations of medical images, and therefore we designed our experiments for the medical domain. We also strongly believe that our method will largely benefit the medical imaging community: as we pointed out, while it may be non-trivial to obtain paired image-text data for arbitrary natural images, paired image-text examples are produced naturally by medical experts in their daily workflow and exist in abundance worldwide. Our paper presents an effective way to utilize these already existing paired data. On the other hand, it is not our intent to claim that our method is only specific or applicable to medical image understanding. It is in fact an interesting future direction to validate whether the same approach will further push the state-of-the-art on general image understanding tasks. We will make this clear in an updated version of this paper.
>
> We hope that these clarifications can help you better understand our work and resolve some of the concerns raised!
>
>
> References:
>
> [1] Lu, et al. "ViLBERT: Pretraining task-agnostic visiolinguistic representations for vision-and-language tasks." In NeurIPS 2019.
>
> [2] Su, et al. "VL-BERT: Pre-training of generic visual-linguistic representations." In ICLR 2020.

---

### Official Review · AnonReviewer2 · 2020-10-28
**An interesting work, utilizing the contrastive learning to learn medical visual representations**

**Rating:** 6
**Confidence:** 4

**Review:**

#####################   Summary   ####################

This paper presents the Contrastive VIsual Representation Learning from Text (ConVIRT) pretraining strategy to learn fine-grained medical visual representations of medical images by pretraining on large-scale image-report pairs. As a result, ConVIRT improves medical visual representations by maximizing the agreement between true image-text pairs versus random pairs via a bidirectional contrastive objective between the image and text modalities. The authors demonstrate that ConVIRT can help improve performance on down-streaming tasks like image classification and image retrieval tasks.


#####################   Strengths   ####################

(1) This paper is really clearly written. The paper is easy to follow and understand.

(2) The paper explores an interesting direction of learning fine-grained medical visual representations for medical image understanding.

(3) The proposed ConVIRT is well motivated. The experimental results are very solid.

#####################   Weakensses   ####################

(1) The paper is limited in its novelty borrowing ideas from some previous works: (i) contrastive learning [1][2] and (ii) image-text representations pretraining [3]. However, it is an interesting idea of applying contrastive learning to medical image and text. So, I think this paper is novel to some extent, but not that novel, since it mainly makes some incremental contribution by combining ideas from existing work.

(2) The analysis is not convinced (see below).

#####################   Questions   ####################

I have some questions for the authors:

(1) What the medical visual representations have learned?

(2) Compared with the baseline methods (section 3.3), why the proposed ConVIRT can achieve the best results? How these results can be achieved?

(3) Why the proposed ConVIRT can learn better medical visual representations than the SimCLR [1] and MoCo v2 [2] under the setting of image-only contrastive learning (section 5)? How they differ from representations learned by existing methods, e.g., SimCLR [1] and MoCo v2 [2]?

(4) Can the medical visual representations pretrained on the chest image be further finetuned on the bony image?

Overall, although the paper doesn't provide insights around what the representations have learned and how they differ from representations learned/used by existing methods, they have provided substantial evidence to suggest that pretraining helps in a lot of downstream tasks. In other words, the proposed ConVIRT seems to be useful for the researchers in the medical AI field.


[1] A simple framework for contrastive learning of visual representations. In ICML 2020.

[2] Improved baselines with momentum contrastive learning. arXiv preprint arXiv:2003.04297.

[3] ViLBERT: Pretraining task-agnostic visiolinguistic representations for vision-and-language tasks. In NeurIPS 2019.

####################  After Rebuttal  ####################

I thank the authors for responding to the comments and have read them carefully. The authors have addressed my concerns in the rebuttal.

---

> ### Author Response · Authors · 2020-11-19
> **Authors' Response to Reviewer 2**
>
> Dear Reviewer 2,
>
> Thank you for your detailed comments, and for kindly recognizing the clear presentation, valuable direction, and solid experimental results in our paper! You are mainly concerned with the limited novelty and the analysis in our paper. We respond to these concerns and questions below:
>
> **Weakness 1** (incremental contributions): We first agree that our method is inspired by recent work on image-only contrastive learning, such as SimCLR [1], as we also made clear in our introduction. We further contribute by extending these methods to a two-modality contrastive setting, and showed that it is notably superior to image-only contrastive methods (Section 5), as well as baselines without contrastive objective. We argue that this contribution is especially important for medical image understanding, where paired text data naturally exists and the extremely high cross-class similarity of images renders image-only contrastive methods less useful (see more details in our response to Question 3). Please also see our **general response** to all reviewers, where we clarify the differences between our work and existing vision-language pretraining work.
>
> **Question 1 & 3** (learned representations & comparisons against image-only contrastive learning): As we have shown, image-only contrastive methods are less effective for learning representations of medical images, mainly due to the high cross-class similarity of such images. Here is an illustrative example: the two medical images shown in Figure 1 of our paper correspond to two drastically different disease categories (cardiomegaly and pleural effusion), yet the structure and characteristics of the two images are largely the same. This is also true for almost all disease categories of the same medical image type. This renders the contrastive objective in image-only methods less effective: while a model can effectively learn by contrasting cropped views from an image of a bird and that of a car, cropped views from a “cardiomegaly” chest X-ray image and an “effusion” chest X-ray are largely similar and do not provide enough information for contrastive learning. Our proposed method ConVIRT instead relies on learning true image-text pairings, therefore is not limited by the high cross-class similarity of medical images. To help understand the differences in representations learned with different methods, we will include more visualizations of encoders learned by SimCLR and our method in an updated version of our paper.
>
> **Question 2** (comparisons with baselines): Compared to random or ImageNet initializations, our method learns in-domain visual features that better capture the characteristics of medical images. Compared to the captioning-based methods based on decoding the paired text, we believe that our method benefits from explicitly contrasting the encoded images and the whole sentence encodings via a cosine similarity in the loss function. In other words, while a captioning-based model can learn to generate high-quality reports with a strong decoder, our pretraining task “forces” the model to focus on discriminating true versus fake paired text using the learned image representations, and via doing so, learn to capture key visual features. Compared to the “Binary-Contrastive” baseline, our method benefits from the effective use of the cosine similarity function and the efficient use of more negative examples in the NCE loss.
>
> **Question 4** (further fine-tuning for different domains): We hypothesize that further fine-tuning medical visual representations pretrained on chest images for other domains will help, since medical images from different domains can sometimes share similar visual characteristics (e.g., fracture of bone structure or the contour of muscle structures). We believe that this will be an interesting idea to validate in future work.
>
> We hope that these clarifications can help you better understand our work and resolve some of the concerns raised!
>
> References:
>
> [1] Chen, et al. "A simple framework for contrastive learning of visual representations." In ICML 2020.

---

### Official Review · AnonReviewer1 · 2020-10-29
**Contrastive Learning of Medical Visual Representations from Paired Images and Text**

**Rating:** 5
**Confidence:** 5

**Review:**

In this work, the authors propose a new model, named ConVIRT to learn the medical visual representation from paired image and textual data in an unsupervised strategy. In ConVIRT, they mainly use a contrastive loss with two modalities (images and texts) as inputs to learn the representation. The experimental results show their proposed model achieve higher performance than other methods in image classification and zero-shot retrieval tasks.

Strength.
In this work, the authors propose a simple and straightforward method to learn the image visual representation using modified contrastive loss with two modalities as inputs. Besides, the experimental comparisons show their model outperforms other baselines and methods.

Weakness.
1. Novelty. In this work, the main contribution is the proposed modified contrastive loss with two modalities as inputs. It is an interesting and challenging problem that how to extract the visual representation in an unsupervised strategy. However, the proposed modified contrastive loss is below the standard of top-tier conference ICLR.
2. Experiment. This work evaluates their model in many tasks and datasets. But, it might miss some baselines or other state-of-the-art methods. (1) Baselines. I suggest the authors could add the baseline using triplet loss. Both contrastive loss and triplet loss could enlarge the distances between different classes. (2) Baselines. I also suggest the authors could add the baseline only using the images as inputs, just like Chen et al. 2020a. and maybe only using the text data as inputs.

3. Implementation details. (1) BERT encoder. In section 2.3, they use a BERT encoder to extract the textural features. I suggest the authors could give more details about it, such as how to initialize the parameters? As we known, there exists a huge gap between contexts form website and that from the medical report, since there are so many specific nouns. (2) Text transformation function. I might not agree with the text transformation function used in this work as the following reasons. There might exist a big gap and different meanings between two sentences even in one document. As the goal is to maximize the agreement between the true image-text representation pairs, there might occur a conflict between one image and two different meaning sentences sampled from the same documents.

4. Visualization. (1) I suggest the authors could give more visualization examples about the learned visual and textual representation, such as t-SNE. Such visualization might also address my above the concerns about the potential conflict during training. In these t-SNE examples, according to the proposed pipeline, the distance between the learned visual representation and the learned textual representation of the corresponding sentences in the same document should be smaller than the distance between the visual representation and some similar sentences from other documents.
5. New type of baselines. This work aims to learn the useful visual representation from both images and texts. However, I suggest that they should report another kind of baselines, only using images to learn the visual representation in an unsurprised way, such as Chen et al. 2020a and [1] (Unsupervised Representation Learning by Predicting Image Rotations, ICLR 2018).
The results of the proposed method should outperform this kind of baseline.

---

> ### Author Response · Authors · 2020-11-19
> **Authors' Response to Reviewer 1**
>
> Dear Reviewer 1,
>
> Thank you for your detailed comments, and for kindly recognizing the simplicity and superior performance of our proposed method! We however would like to respond to the comments about weaknesses, some of which we have also addressed in the original submission. Please see our responses below:
>
> **Weakness 1** (novelty): We agree with the reviewer that our main technical contribution is the cross-modality contrastive learning framework; we however disagree that this does not constitute a novel contribution to a conference like ICLR. Our ConVIRT method, despite its simplicity, was original and not used for learning image representations in previous work. Please also see our **general response** to all reviewers in which we clarify the contributions of our work and its differences with existing vision-language pretraining work.
>
> **Weakness 2.1** (comparison against triplet loss): We focused our experiments on comparing against caption-based methods, binary contrastive loss, and existing image-only contrastive methods. We did not include a comparison with triplet loss mainly because a similar triplet loss was already shown to be inferior to the NCE-based loss in the SimCLR paper [1] (Section 5.1 of their paper), which our paper was inspired by.
>
> **Weakness 2.2** (comparison against image-only contrastive methods such as Chen et al., 2020a): We already showed comparisons against image-only methods in Section 5, Table 2 of our original submission. In addition to SimCLR, we also compared against the more effective MoCo v2 [2] method, and showed that our ConVIRT pretraining outperformed both methods on our classification and retrieval tasks (+4.1 on RSNA classification, +4.6 on CheXpert classification, and +22.3 on image retrieval compared to MoCo v2).
>
> **Weakness 3.1** (BERT encoder): Due to space constraints, we included details of our BERT encoder in Appendix A of our original submission. In short, we used the ClinicalBERT [3] in our implementation, which was pretrained on a large corpus of clinical text and was shown to achieve state-of-the-art performance on several clinical language understanding tasks. We further fine-tuned the BERT encoder in our pretraining to help the encoder better adapt to our task. For clarity, we will include more details about the text encoder in the main text in an updated version.
>
> **Weakness 3.2** (concern on text transformation): We agree with the reviewer that in our setting, different sentences in the same document may have different semantic meanings. We however note that this does not lead to an ineffective objective for our pretraining task. Our cross-modality NCE objective aims at learning image representations that are close to all the positive sentence representations. In the case where an image is paired with multiple sentences, we argue that our training method is still effective in positioning the image representation closer to each text representation in a subspace of the entire representation space. The end result is that an image vector is close to each of its positive paired sentences in a specific subspace, while still being distant from all negative sentences. Our linear classification experiments (Section 4.1) corroborate this: our pretraining has led to image representations that are substantially better at discriminating all types of labels that often occur in different sentences of the report (e.g., pneumonia, effusion, etc.) even without any fine-tuning, highlighting that conflicting sentence meanings did not hurt the validity of our training process.
>
> **Weakness 4.1** (visualization): We did include comparisons of t-SNE visualizations of the learned image representations in Appendix F of the original submission. The t-SNE visualizations have shown that our pretraining has led to notably better clustering of medical images from different categories, despite that no label is used in pretraining. For clarity, we will move our t-SNE visualizations to the main text. We further note that our zero-shot text-image retrieval task (Section 4.2) was designed for the exact purpose of understanding the distance of the image and text representations before/after the pretraining, and we showed that the pretraining task indeed helped the image representations to be positioned closer to the correct text queries.
>
> **Weakness 5** (new baselines of image-only contrastive learning): We have already compared against methods of this category, i.e., SimCLR [1] and MoCo v2 [2] in Section 5 of our original submission, and showed that our method substantially outperformed these baselines. Please also see our response to Weakness 2.2 for more details.
>
> We hope that these clarifications can help resolve some of the concerns raised.
>
> References:
>
> [1] Chen, et al. "A simple framework for contrastive learning of visual representations." In ICML 2020.
>
> [2] Chen, et al. "Improved baselines with momentum contrastive learning." arXiv preprint arXiv:2003.04297 (2020).

---

### Author Response · Authors · 2020-11-19
**General Response to All Reviewers**

Dear reviewers,

Thank you for your detailed comments on our work! We are especially encouraged by all of your kind comments on the simplicity of our proposed ConVIRT method, its superior performance on medical imaging tasks, and the clear presentation of our paper!

A common concern raised in the comments is the novelty of our work and its connection to existing vision-language pretraining work such as ViLBERT [1]. We would like to address this concern in this general response.

We first agree with the reviewers that our main technical contribution is the proposed cross-modality contrastive learning framework ConVIRT, and that the aim of ConVIRT is in line with that of vision-language pretraining work such as ViLBERT. We however want to argue that our work differs from existing vision-language pretraining work in two crucial aspects.

First, existing work such as ViLBERT pretrains image encoders with a binary classification task, in which a trainable linear layer is added on top of the combined image and text vectors to predict whether an image-text pair is positive or negative (see the end of Section 2.2 of the ViLBERT paper). This differs from our method which uses two cross-modality NCE-based losses based on cosine similarity between the transformed image and text vectors. To show that our method in fact outperformed the binary classification method used in existing work, in our paper we compared ConVIRT with a similar binary pretraining baseline (the “Contrastive-Binary” baseline in our Section 3.3, and results in Table 1). We showed that on all classification and retrieval tasks, our NCE-based training objectives (Eq. 2 & 3) are notably more effective than this binary pretraining baseline. We believe that this represents a distinct and novel contribution of our work.

Second, existing vision-language pretraining such as ViLBERT requires segmented image regions as input (see Section 2.2 of the ViLBERT paper). This is an important step that helps existing methods achieve excellent performance, but at the same time renders them inapplicable to medical images where image segmentation tools are extremely scarce and need to be produced for each anatomic structure and imaging modality separately. In contrast, we showed that this step is not necessary to learn good representations of medical images. Via the NCE-based losses, ConVIRT can use the entire (transformed) image as input, and learn very fine-grained representations required by medical imaging tasks. We argue that this simpler approach represents a valuable contribution that renders ConVIRT more practically applicable and useful for medical image understanding tasks, as also pointed out by Reviewer 2.

In addition to these important differences with existing work, we believe that our work represents the first systematic attempt at comparing different initialization strategies for medical imaging, an important domain for vision research. We also collected and will make available new image-image and text-image retrieval datasets that can facilitate future research in this direction. Altogether we believe that our work represents a substantial and novel contribution to ICLR. We hope that these clarifications can help the reviewers better understand our work and resolve the concerns!

References:

[1] Lu, et al. "ViLBERT: Pretraining task-agnostic visiolinguistic representations for vision-and-language tasks." In NeurIPS 2019.

---

### Author Response · Authors · 2020-11-25
**Update to our PDF submission**

Dear reviewers,

We have updated our PDF submission by incorporating suggestions from all of you. Our main changes in this update include:

1. **BERT encoder**: For clarity we have moved more details on our BERT encoder from the appendix to the main text (Section 2.3 second paragraph), following the feedback from Reviewer 1.

2. **t-SNE visualizations**: We have moved our t-SNE visualization figures and the accompanying explanations from the appendix to the main text (Section 4.2 second paragraph), following the suggestion from Reviewer 1.

3. **Model visualizations and explanations**: We have added new saliency map visualizations on sampled medical images for different pretraining methods, including ImageNet, SimCLR, MoCo and our proposed ConVIRT model. We show these visualizations in Figure 4 and explain them in the second paragraph of Section 5. This follows suggestions from Reviewer 2.

4. **Related work**: We have improved our comparisons to existing vision-language pretraining work in the related work section (Section 6). We have added more details to clarify the differences between our work and existing work, and corrected some potentially misleading claims as pointed out by Reviewer 3.

We want to sincerely thank all reviewers again for your great suggestions that help improve this work!

---

### Decision · Program_Chairs · 2021-01-07
**Final Decision**

**Decision:**

Reject

**Comment:**

The proposed ConVIRT learns representations of medical data from paired image and text data.
While the paper addresses a relevant problem, the reviewers agree that the method has limited novelty. Two reviewers find and that the experiments are not convincing. One reviewer notes that the paper does not compare to the state-of-the-art methods for the tasks.